# The Driving Mechanism of Urban Land Green Use Efficiency in China Based on the EBM Model with Undesirable Outputs and the Spatial Dubin Model

**DOI:** 10.3390/ijerph191710748

**Published:** 2022-08-29

**Authors:** Liangen Zeng

**Affiliations:** College of Urban and Environmental Sciences, Peking University, Beijing 100871, China; zengliangen@pku.edu.cn

**Keywords:** urban land green use efficiency, EBM model with undesirable outputs, spatial durbin model

## Abstract

Green development is necessary for building a high-quality modern economic system. The contribution of the article mainly includes the following three parts: First is the study on the urban land green use efficiency (ULGUE) in 30 provinces of China from 2008 to 2018 by adopting the epsilon-based measure (EBM) model with undesirable outputs to yield a more accurate and reasonable assessment result. In addition, the spatial agglomeration characteristics were analysed according to the spatial autocorrelation analysis. Thirdly, the spatial Durbin model was applied to analyse the driving factors of the WRGUE, which considers the spatial effects. The findings are as follows: (1) The regional differences in ULGUE were very significant, with the number decreasing from the coastal region to inland. (2) ULGUE showed a significantly positive spatial autocorrelation, and the spatial homogeneity was more significant than the spatial heterogeneity for ULGUE. (3) Economic development level, technical progress level, and urban population density have a significant impact on ULGUE, while the higher the proportion of the secondary industry in GDP, the lower the level of ULGUE. The research results may be a useful reference point for policymakers.

## 1. Introduction

Urban land is not only the carrier of urban economic activities, but also the basic living space of urban residents [1]. Rapid urbanisation seriously changed the Earth’s surface due to natural surface transformation [2]. The globe is at the stage of rapid urbanisation with the dividends of economic growth, employment increase, and improvement in living standards. Still, it also faces many problems, such as the excessive use of biological resources, causing negative environmental impacts. In such a context, green development gradually became an effective method to ease the double pressure of economic development and resource shortage, which considers the restrain imposed by limited resources and pollution of the environment.

Since the opening up of the economy, the pace of urbanisation in China quickened continuously. The urbanisation level rose from 17.9% in 1978 to 64.7% in 2021 [3]. Although the urban construction in China is enormous, there are problems at the same time. First, urban construction in China is consuming a lot of land resources and energy, and it is causing serious environmental pollution. In addition, urban regions in China are consuming large quantities of energy and emitting large volumes of pollutants. As Figure 1 shows, the urban construction land area in China increased from 10,816.5 km^2^ in 1987 to 56,075.9 km^2^ in 2018 [1]. Available land for urban construction is becoming scarce, and this constrains the development of urbanisation. Therefore, improving urban land use efficiency is necessary for the Chinese government to tackle the human–land contradiction, and to ensure the sustainable utilisation of urban land.

Urban land use efficiency is one of the main indicators used to measure the land output capacity and development quality of urban areas [4]. The main calculation methods of ULUE are stochastic frontier analysis (SFA) and data envelopment analysis (DEA). SFA is a parametric method for calculating the technical efficiency of the decision-making unit, which considers the influence of random factors on efficiency [5]. The DEA method is non-parametric, and it can deal with complex systems that have multiple inputs and outputs [6]. The traditional ULUE only considers the economic and social scope of land use [7], and it ignores the green concept of the coordinated development of economic, social and environmental factors [8]. In recent years, many scholars explored the undesirable output factors in the index system of ULUE measurement, either with SFA [9,10,11,12] or DEA [4,13,14,15,16,17]. SFA is the requirement for prior assumptions of functional form. Thus, inaccurate results may occur because of incorrect functional forms. DEA does not need any prior specification on the functional form of the frontier. Hence, DEA methods are more widely used for ULUE measurements than SFA methods. However, there is no explicit conception of the comprehensive ULUE that takes account of undesirable output factors. With the wide spread of green development concepts, many scholars put forward the concept of urban land green use efficiency (ULGUE), which pursues more economic outputs based on less resource consumption and pollutant emission from urban land [8]. Yan and Wang [18], and Wang et al. [19] all applied DEA methods to calculate ULGUE.

In summary, the existing research had some successes, and the main measurement methods are DEA methods, which can be classified into two types: radial models, such as CCR or BCC, and non-radial models, such as SBM. The SBM model directly captures the non-radial slacks that are not considered in the radial models, which may cause the loss of the original proportionality [20,21,22,23]. Therefore, the SBM became the main DEA method for calculating ULUE. However, SBM still has some shortcomings. One such shortcoming is that the linear proportion information between input variables is not considered, so the efficiency scores of the decision-making unit may be underestimated [24,25]. To overcome the deficiencies of the SBM method, Tone and Tsutsui [26] proposed an EBM DEA model, which combines the advantages of the traditional DEA model and the SBM model, and is compatible with radial and non-radial mixed distance functions. Therefore, the paper applied the EBM model with undesirable outputs to calculate the ULGUE of 30 provinces in China from 2008 to 2018. This led to more reasonable results for ULGUE.

This paper evaluated the ULGUE of 30 provincial administrative regions in China from 2008 to 2018, and then analysed its spatiotemporal characteristics and its driving factors. The main contributions of this paper are: (1) Applying a more precise method for measuring ULGUE. An epsilon-based measure (EBM) model with undesirable outputs was applied to evaluate the ULGUE, which makes up for the defects of SBM methods and can get more reasonable results of efficiency calculation. (2) The spatiotemporal characteristics of ULGUE in the country, as well as the eight regions, were carefully analysed; the spatial agglomeration characteristics of ULGUE of the 30 provinces were comprehensively analysed according to Moran’s I. Therefore, we can get an overall, transparent and in-depth understanding of the green development level of urban land from the paper. (3) The spatial Dubin model (SDM) was used for an empirical analysis of the influencing factors of ULGUE. The empirical research results can be used as a reference in promoting ULGUE and formulating the policies related to land and space governance.

The structure of the article is: The methodology is shown in Section 2. The Results and Discussion are in Section 3, which analyses the characteristics of ULGUE in China and discusses the influence factors of ULGUE. Section 4 summarises the conclusions, gives policy suggestions, and proposes some future work.

## 2. Methodology

### 2.1. Research Area and Framework

This study selected the ULGUE of 30 provincial administrative regions in the Chinese Mainland from 2008 to 2018 as the main research object (the data for Tibet had significant missing material and were eliminated), and analysed its temporal and spatial distribution properties and influencing factors. The provinces are divided into eight regions based on the National Bureau of Statistic of China (Figure 2). The analysis framework of this paper is presented in Figure 1.

### 2.2. The Conceptual Framework of ULGUE

Based on comprehensive and scientific principles, ULGUE is defined as the comprehensive mapping of an urban land production system, seeking more economic outputs and less environmental impacts in a condition of stable or decreasing inputs of productive factors [27]. The core of ULGUE is to pursue the coordination and unity of economic growth, resources, and the environment in the urban land production system. Therefore, the indicator selection for the ULGUE calculation considers not only the economic benefits of urban land, but also the social and environmental benefits. In reference to existing studies of Ge et al. [4], Liu et al. [13], Zhang et al. [14], Song et al. [15], Jiang [16], and Wu et al. [17], this paper selected urban capital stock, urban employees and urban construction land as the input indicators, the added value of the second and the third industries as the desired output, and the CO_2_ emissions from the second and the third industries as the undesired output (Table 1).

Urban capital stock. This paper used the equation *K_it_* = *I* + (1 − *δ*)*K_it−_*_1_, where *K* and *I* are the capital stock and the urban investment in fixed assets; *i* and *t* stand for the province and year; and *δ* represents the depreciation rate. According to Zhang et al. [28], *δ* is 9.6%. The urban capital stock in 2008 was equal to the value of the urban investment in fixed assets in 2008 divided by 10%. The data of the urban investment in fixed assets were converted to the 2008 constant price and came from the NBSC [1].

Urban employees. This paper selected employees of the second and the third industries as the urban employees. The data were gathered from the statistical yearbooks of the Chinese provinces (2009–2019) [1].

Urban construction land. This paper chose the urban construction land of each province as one of the input indicators. The data came from NBSC [1].

The added value of the second and the third industries: To diminish the impact of inflation, the added value of the second and the third industries were converted to the 2008 constant price. The data came from *China Statistical Yearbook (CSY)* (2009–2019) [29].

CO_2_ emissions from the second and the third industries: the date came from the China Emission Accounts and Datasets (https://www.ceads.net/data/province/, accessed on 22 August 2022).

### 2.3. The Influential Mechanism of ULGUE

In reference to the relevant literature, this study selected economic development level, government regulation, industrial structure, technical progress level, urban population density, and opening up level as the independent variables (Table 2). The data came from NBSC [1] and CSY (2009–2019) [29].

#### 2.3.1. Economic Development Level

Economic development level determines the number of input elements per unit land area that can affect urban land use. If the region has a higher level of economic development, the more high-quality production factors would be attracted to the region, which may improve the ULGUE [30]. Based on the studies of Zhong et al. [8], Song et al. [15], Yang and Wang [18], Zeng et al. [31], Jiang et al. [32], and Yu and Su [33], the level of economic development was measured by using per-capita GDP.

#### 2.3.2. Government Regulation

Government regulation affects urban land use. Government financial expenditure will affect the relevant decisions of urban land use, and change the specific land use behaviour of land. In the economic growth-oriented performance appraisal system, local governments mainly formulate land requisition policies to expand urban land areas to alleviate problems, such as the shortage of local financial resources [30]. Based on Lu et al. [6], Ge et al. [14], Lu et al. [30], Fan et al. [34], Tu et al. [35], and Ge and Liu [36], the proportion of government expenditure in GDP was selected to measure government regulation.

#### 2.3.3. Industrial Structure

The development of industry depends on land, and different industrial structures will have different impacts on land use mode. Generally speaking, second industry is the main source of CO_2_ emissions. Based on Ge et al. [14] and Jiang et al. [32], Ge and Liu [36], and Gao et al. [37], the proportion of secondary industry in GDP was selected to measure the industrial structure.

#### 2.3.4. Technical Progress Level

Technical progress can effectively promote the use level of land, as well as change the mode of urban land use and improve ULGUE. Based on the existing research of Lu et al. [6], Zhong et al. [8], Yu et al. [38], and Chang and Chen [39], this study selected technical progress level as the independent variable.

#### 2.3.5. Urban Population Density

The increase in urban population density will bring capital, technology, labour, and other production factors to the city, which brings about scale economies and promotes technological efficiency. However, when the urban population density is too high, it causes congestion and leads to the destruction of the ecological environment. Based on Wang et al. [2], Lu et al. [6], Song et al. [15], Jiang et al. [32], and Yang et al. [40], urban population density was selected as an important dependent variable.

#### 2.3.6. Opening up Level

Theoretically, if the region has higher levels of openness, the region will be more likely to enjoy the fruits of knowledge and technology spillover from overseas. Based on Jiang et al. [32] and Yu and Su [33], this study selected the foreign trade volume as a proportion of total GDP to measure the opening-up degree of each city.

**Table 2 ijerph-19-10748-t002:** ULGUE driving factors.

Explanatory Variable	Variable Definition and Unit	References	Expectation
Economic development level	Per capita GDP (10^4^ RMB)	[8,15,18,31,32,33]	Positive
Governmental regulation	Proportion of fiscal expenditure in GDP (%)	[6,14,30,34,35,36]	Unknown
Industrial structure	Proportion of the added value of the secondary industry in GDP (%)	[14,32,36,37]	Positive
Technical progress level	Proportion of the R&D expenditure in GDP (%)	[6,8,38,39]	Negative
Urban population density	Urban resident population per square kilometre (person/sq. km)	[2,6,15,32,40]	Unknown
Opening up level	Proportion of foreign trade in GDP (%)	[32,33]	Positive

### 2.4. The EBM Model with Undesirable Outputs

In this paper, the EBM model with undesirable outputs was used to measure the ULGUE, which not only combines the advantages of both radial and non-radial DEA methods, but also can deal with the undesirable outputs. The EBM model with undesirable outputs is indicated in Formula (1) [41]:


(1)
$$\begin{array}{l} T^{*}=\min\left[\frac{\theta -\varepsilon_{x}\sum_{i=1}^{m}\frac{w_{i}^{-}s_{i}^{-}}{x_{\mathrm{io}}}}{\varphi +\varepsilon_{x}\sum_{r=1}^{s}\frac{w_{r}^{+}s_{r}^{+}}{y_{\mathrm{ro}}}+\varepsilon_{b}\sum_{p=1}^{q}\frac{w_{p}^{-}s_{p}^{-}}{b_{\mathrm{po}}}}\right]\\ s.t\left\{\begin{array}{c} \sum_{j=1}^{n}x_{\mathrm{ij}}\lambda_{j}+s_{i}^{-}={\theta x}_{\mathrm{io}}\;\;\;\;\;\;\;\;\;\begin{array}{c} i=1,2,\ldots ,m \end{array}\\ \sum_{j=1}^{n}y_{\mathrm{rj}}\lambda_{j}-s_{r}^{+}={\varphi y}_{\mathrm{ro}}\;\;\;\;\;\;\;\;\begin{array}{c} r=1,2,\ldots ,s \end{array}\\ \sum_{j=1}^{q}b_{\mathrm{pj}}\lambda_{j}+s_{p}^{-}={\varphi b}_{\mathrm{po}}\;\;\;\;\;\;\;\;\begin{array}{c} p=1,2,\ldots ,q \end{array}\\ \lambda_{j}\geq {0,s}_{i}^{-}\geq {0,s}_{r}^{+}\geq {0,s}_{p}^{-}\geq 0 \end{array}\right. \end{array}$$


In where *T** indicates the technical efficiency value of the the decision-making unit *o* (0 < T ≤ 1); *n* is the number of the decision-making units; *m*, *s* and *q* represent the inputs, desirable outputs, and undesirable outputs, respectively; *θ* and *φ* stand for the radial programming parameters; λ is the intensity vector; $w_{i}^{-}$, $w_{r}^{+}$, and $w_{p}^{-}$ represent the weights of the input, desirable output and undesirable output variables, respectively, and satisfy $\sum_{i}^{m}w_{i}^{-}=1$, $\sum_{r}^{s}w_{r}^{+}=1$, and $\sum_{p}^{q}w_{p}^{-}=1$ ($w_{i}^{-}\geq 0,w_{r}^{+}\geq 0,w_{p}^{-}\geq 0,\forall m,s,q$); $s_{i}^{-}$, $s_{r}^{+}$, and $s_{p}^{-}$ are the slacks of the input *i,* desirable output *r* and undesirable output *p*, respectively, and *ε* indicates the importance of the non-radial measure efficiency and is in the range of [0, 1].

### 2.5. Spatial Correlation Analysis

In this paper, Moran’s I is applied to analyse the spatial autocorrelation of ULGUE among provinces in China. Spatial autocorrelation refers to the mutual dependence of spatial element attributes on spatial location and is a measure of the spatial agglomeration degree. Global Moran’s I can examine the spatial dependence of ULGUE in the whole region, which is as follows:(2)$$Global\;Moran’I\begin{array}{c} =\frac{\sum_{i=1}^{N}\sum_{j=1}^{N}\mathrm{Wi},j\left({\mathrm{ULGUE}}_{i,t}-{\bar{\mathrm{ULGUE}}}_{t}\right)\left({\mathrm{ULGUE}}_{j,t}-{\bar{\mathrm{ULGUE}}}_{t}\right)}{\left[\frac{1}{N}\sum_{i=1}^{N}{\left({\mathrm{ULGUE}}_{i,t}-{\bar{\mathrm{ULGUE}}}_{t}\right)}^{2}\right]\sum_{i=1}^{N}\sum_{j=1}^{N}W_{i,j}} \end{array}$$

In Formula (2), i and j stand for province i and province j, respectively; n is the number of provinces studied; and in this paper, n = 30. W_ij_ stands for the spatial weight matrix between province i and province j. If province i is adjacent to province j, W_ij_ = 1, otherwise W_ij_ = 0. $\bar{\mathrm{ULGUE}}$ is the average value of the ULGUE. The value range of Global Moran’s I is [−1, 1]. If the value is larger than 0, it means that there is a positive spatial dependence for ULGUE, while the value less than 0 indicates a negative spatial autocorrelation.

The Global Moran’s *I* can reflect the overall spatial correlation of ULGUE, but some local features may be ignored. The Local Moran’s *I* can test the degree of clustering or dispersion in local regions, which is calculated as:(3)$$\mathrm{Local}\mathrm{Moran}’I\begin{array}{c} =\frac{N\left({\mathrm{ULGUE}}_{i,t}-{\bar{\mathrm{ULGUE}}}_{t}\right)\sum_{j=1}^{N}W_{i,j}\left({\mathrm{ULGUE}}_{j,t}-{\bar{\mathrm{ULGUE}}}_{t}\right)}{\sum_{i=1}^{N}{\left({\mathrm{ULGUE}}_{i,t}-{\bar{\mathrm{ULGUE}}}_{t}\right)}^{2}} \end{array}$$

The local Moran’s *I* usually can be represented by the Moran scatter plot (MSP) map and local indicators of spatial association (LISA) map. The MSP and LISA maps are divided into four quadrants: the first quadrant is the high–high (HH) agglomeration area, which means that the province with a high ULGUE value is close to other provinces with high ULGUE values. The second quadrant is the low–high (LH) agglomeration area, indicating that the province has a low ULGUE value but is surrounded by provinces with high ULGUE values. The third quadrant is the low–low (LL) agglomeration area, indicating that the province with a low ULGUE value is surrounded by other provinces with low ULGUE values. The fourth quadrant is the high–low (HL) agglomeration area, which implies that a province has a high ULGUE value, but is surrounded by provinces with low ULGUE values [42,43].

### 2.6. Spatial Durbin Model

When there is a strong spatial relationship with the dependent variable, the spatial econometric model should be introduced, which makes the parameter estimation results more convincing. There are three classical spatial econometric models: the spatial lag model (SLM), the spatial error model (SEM), and the spatial Dubin model (SDM). In this section, we bring in these three models, and then select the most proper one by Wald and likelihood ratio tests.

The SLM contains endogenous interaction effects among the explained variables [25], indicating that the spatial autocorrelation can be explained by a spatially lagged explained variable [22]. The equation of the SLM is as follows:Y = ρWY + βX + ε ε ~ N(0, σ^2^ I_n_)(4)

In Equation (4), Y is the explained variable; X indicates the explanatory variable; ρ is the spatial autoregression coefficient; β indicates the regressive spatial coefficients; W represents the spatial weight matrix; and ε indicates a random error term.

The SEM contains interaction effects among the error terms, implying that an explained variable is randomly affected by adjacent areas [22]. The expression of the SEM is as follows:Y = βX +μ μ = λWX + ε ε ~ N(0, σ^2^ I_n_)(5)

In Equation (5), µ denotes the random error vector. Other variables in Equation (5) are consistent with those in Equation (4).

The SDM considers the spatial effects caused by explained variables or explanatory variables or error terms well [44,45,46], and it can obtain more accurate regression coefficient estimates [47], so it is more widely used than SEM or SLM. Therefore, the SDM was applied to analyse the driving factors of ULGUE in China. The SDM can be expressed as:Y = ρWY + βX + θWX + ε ε ~ N(0, σ^2^ I_n_)(6)

In Equation (6), *θ* stands for the spatial regressive coefficients; in the same vein, the variables in Equation (6) have the matching definition in Equation (4).

Wald and likelihood ratio tests were carried out to assess whether SDM could be degraded to SEM or SLM. As Table 3 shows, the test results all reject the null hypothesis that SDM can be simplified into SEM or SLM at the significance level of 1%.

## 3. Results and Discussion

### 3.1. National Characteristics of ULGUE

The ULGUEs of 30 provinces in China from 2008 to 2018 were calculated and are shown in Table 4 and Figure 3. From the national perspective, the annual mean value of ULGUE in China’s 30 provinces was 0.759 during the study period. The national level of ULGUE fluctuated between 0.722 and 0.78, with the highest year being 2013 and the lowest year being 2009. Figure 3 shows that there were significant spatial differences in ULGUE decreasing from the east coast to the inland west. Figure 4 gives the evolutionary trends of UGLUE and shows that the level of ULGUE in coastal provinces is generally high, while that in the Northeast and Northwest regions is generally low (Table 5).

### 3.2. The Regional Characteristics of ULGUE

#### 3.2.1. Northern Coast

The annual mean value of ULGUE in the northern coast is 0.931, which is at a high level. The northern coast is the economic centre of northern China. The Beijing–Tianjin–Hebei urban agglomeration and the Shandong Peninsula urban agglomeration in the region reached the mature stage, and the levels of urban land use and emission reduction are high. The region is the political, cultural, international communication, and innovative center of China [48]. Beijing was on the production frontier surface during the research period, with Tianjin, Shandong and Hebei behind. The deregulation of Beijing’s non-capital functions relocated many manufacturing industries in the urban area into Hebei, affecting the ULGUE of Hebei.

#### 3.2.2. Eastern Coast

The eastern coast has the highest level of ULGUE of the eight regions. It contains the most dynamic economic areas, it is at the forefront of opening up in China, and it has the strongest innovation capability in China [49], along with a high level of land use technology. Shanghai was on the production frontier surface during the research period, and the annual mean values of Zhejiang and Jiangsu were 0.958 and 0.948, so the difference between these two provinces is small.

#### 3.2.3. Southern Coast

The annual mean value of ULGUE on the southern coast was 0.834, which is a high level. The southern coast is one of the most developed areas for light industry in China, and it was also the first to carry out reform and to open up. It benefited from the spillover of land use technology and management experience from abroad, which promotes ULGUE. Fujian was on the production frontier surface during the research period, while the ULGUE level of Hainan was relatively low.

#### 3.2.4. Northeast

Northeast China has the lowest level among the eight regions. Northeast China is China’s traditional old industrial base, as its industrial structure is dominated by heavy industry with large urban land pollution. There was both continuous population outflows and the lack of vitality in economic growth in northeast China in recent years [50], which seriously inhibited regional development. The annual mean values of ULGUE in Liaoning, Jilin, and Heilongjiang are 0.656, 0.468, and 0.566, respectively, showing significant differences.

#### 3.2.5. Middle Yellow River

The overall level of ULGUE in the Middle Yellow River is slightly higher than the national average level. It is worth noting that the regional ULGUE level shows a downward trend since 2011. Middle Yellow River is an important base of the energy and chemical industries in China, and the regional industrial structure has the characteristics of resource-intensive and labour-intensive industries, so it is difficult to reduce emissions. In the region, the ULGUE in Henan and Shaanxi are at a relatively high level, while that of Inner Mongolia is low.

#### 3.2.6. Middle Yangtze River

Compared to the national average level, the ULGUE of Middle Yangtze River is slightly low. The region is an important energy, raw material, and equipment manufacturing base in China, and has a comprehensive transportation hub. Regional economic development highly depends on resources, and has the characteristic of a high proportion of low-level industries with high consumption of energy and resources [51]. After the government implemented the central rise strategy, a number of industrial parks were established in the Middle Yangtze River, and they took many high-emission industries away from coastal areas, affecting ULGUE. In the region, Hunan has the highest level of ULGUE. The ULGUE level of the other three provinces is similar.

#### 3.2.7. Southwest

The annual mean value of ULGUE during the study period in southwest China was 0.706. In the region, only Sichuan (0.795) exceeded the national average level, and the other three provinces were less than the mean level of the whole country. In recent decades, the population growth and economic development in southwest China led to large-scale urban expansion, and the regional urban resident population increased by 49.7% from 2008 to 2018, which is higher than that of China as a whole (38.5%) in the same period [1]. This rapid urbanisation caused many negative effects on regional ULGUE.

#### 3.2.8. Northwest

The annual mean value of ULGUE in each province during the study period in northwest China was only 0.574, which is lower than average in China. From the perspective of provinces, the average annual values in Gansu, Qinghai, Ningxia, and Xinjiang were 0.566, 0.657, 0.535, and 0.536, respectively, which are far lower than the national average. Due to their remote geographical locations and poor natural environments, the technical accumulation and economic output of urban land in northwest China are low, resulting in a low level of ULGUE.

### 3.3. Spatial Autocorrelation Analysis of ULGUE

The Global Moran’s I was used to test spatial autocorrelation in the whole region. The results are in Table 6. During this period, the values of the Global Moran’s I are greater than 0, and are significant at the inspection level, suggesting that China’s ULGLE shows a trend of agglomeration among the provinces. Therefore, we concluded that spatial distribution of ULGLE in China was not random, but a spatial agglomeration pattern over space, and that the spatial econometric model should be applied to analyse the influencing factors of ULGLE.

In this article, the MSP and LISA maps were applied to analyse the local spatial autocorrelation. The acronyms of 30 provinces in China are shown in Table 7. Figure 5, Figure 6 and Figure 7 present the MSPs of ULGLE in China in 2008, 2013, and 2018, respectively. In 2008, 2013, and 2018, most provinces were in the first and third quadrants, and only a few provinces were in the second and fourth quadrants, indicating that the positive spatial correlations of ULGLE above or below the average value were very obvious. However, in 2018, the number of provinces in the second quadrant increased significantly, indicating that the spatial heterogeneity had a certain degree of increase.

In addition, the MSPs in 2008, 2013, and 2018 indicate that many provinces also have relatively fixed positions. Specifically, Beijing, Tianjin, Hebei, Shanghai, Jiangsu, Zhejiang, Fujian, Shandong, and Henan always belonged to the first quadrant (HH agglomeration area), implying that these nine provinces had high ULGLE and were surrounded by provinces with relatively high ULGLE; these provinces are mainly located in eastern coastal China. Anhui, Jiangxi, Hubei, and Hainan were always in the second quadrant (LH agglomeration area), indicating that these four provinces had low ULGLE and were surrounded by provinces with relatively high ULGLE; these provinces are mainly located in central China. Inner Mongolia, Liaoning, Jilin, Heilongjiang, Yunnan, Gansu, Qinghai, Ningxia, and Xinjiang were always in the third quadrant (LL agglomeration area), implying that these nine provinces had low ULGLE and were surrounded by provinces with relatively low ULGLE; these provinces are mainly in northeast and western China. Hunan and Guangdong were always in the fourth quadrant (HL agglomeration area), indicating that these two provinces had high ULGLE and were surrounded by provinces with relatively low ULGLE.

MSP maps cannot show the significance of ULGLE in each province. LISA maps can solve this problem. Figure 8, Figure 9 and Figure 10 depict provinces with significant locations colour coded by different types of LISA coefficients of ULGLE in China. The red, pink, blue, and yellow areas represent the HH, LH, LL, and HL agglomeration areas at the significance level (*p* ≤ 0.05), respectively. As the figures show, the provinces at the significance level were in the HH, LH, and the LL agglomeration areas.

In 2008, nine provinces passed the significance level test. Beijing, Shanghai, Jiangsu, Zhejiang, and Fujian belonged to the HH aggregation area, Hainan was in the LH aggregation area, and Jilin, Heilongjiang, and Gansu were in the LL aggregation area. In 2013, there were eight provinces at the significance level. Tianjin, Beijing, Shanghai, and Fujian were in the HH aggregation area, Hainan stayed in the LH aggregation area, and Xinjiang, Jilin, and Heilongjiang belonged to the LL aggregation area. In 2018, there were five provinces at the significance level. Beijing, Tianjin, and Shanghai were in the HH aggregation area, and Gansu and Xinjiang were in the LL aggregation area. Therefore, China’s ULGLE formed an HH agglomeration area centred on Beijing and Shanghai in 2008, 2013, and 2018.

### 3.4. Empirical Analysis

To avoid the problem of multicollinearity among variables, this paper conducted Pearson correlation analysis and variance inflation factor (VIF) tests. As Table 8 shows, some correlation coefficients between variables were greater than 0.6, indicating that there may be multicollinearity among the variables. The results of the VIF test (Table 9) show that all the values of VIF were less than 5, indicating that severe multicollinearity does not exist in this model [51]. This provided a basis for the next regression analysis.

The Hausman test was applied to test which of the two effects (fixed or random) was the most appropriate for the panel data. The statistical value of the Hausman test was 33.21 (*p* < 0.01), so the fixed effect is the proper approach. The specific expression of the SDM with fixed effects is as follows:LnULGUE_i,t_ = ρW*LnULGUE_i,t_ + β_1_LnDEL_i,t_ + β_2_LnGR_i,t_ + β_3_LnIS_i,t_ + β_4_LnTPL_i,t_ + β_5_LnUPD_i,t_ + β_6_LnOUP_i,t_ + θ_1_W*LnDEL_i,t_ + θ_2_W*LnGR_i,t_ + θ_3_W*LnIS_i,t_ + θ_4_W*LnTPL_i,t_
+ θ_5_W*LnUPD_i,t_ + θ_6_W*LnOUP_i,t_ + ε_i,t_ ε_i,t_ ~ N(0, σ^2^_i,t_ I_n_)(7)
where EDL, GR, IS, TPL, UPD, and OUL express economic development level, government regulation, industrial structure, technical progress level, urban population density, and opening up level, respectively. The SDM with fixed effects contains spatial-fixed effects, time-fixed effects, and spatial- and time-fixed effects. The value of log likelihood in Table 10 suggests that the spatial- and time-fixed effects (log likelihood value = 504.565) had a better fit than the other two effects (Log likelihood values = 289.278 and 510.484). Therefore, it is most reasonable to apply the SDM with spatial and time-fixed effects to empirical analysis.

ULGUE is affected positively and significantly by economic development level (*p* < 0.01), which is consistent with our expectations. Contemporary China is in a transitional period from the primary stage of economic development to the advanced stage. China’s government made use of different kinds of measures, such as fiscal and tax policy, industrial policy, environmental regulation policy, land policy and so on, to give priority to the development of services, high-tech industries, and other industries with high economic outputs and low environmental impacts, which improved the ULGUE.

The estimated coefficient of government regulation is negative, but this was not apparent. The exploitation of land resources are indispensable for local officials in the management of economic growth overall [52]. From 2008 to 2018, the proportion of government expenditure to GDP in China grew from 19.6% to 24%. The Chinese government obviously adopted an active fiscal policy to boost the economic growth, which gave impetus to the expansion of urban land in China. As the economy enters a “new normal” and growth slows, the government expenditure structure needs to be optimised to improve the efficiency of fiscal expenditure.

The industrial structure affects ULGUE markedly (*p* < 0.01), meaning that the increase in secondary industry is inhibiting the ULGUE, which is consistent with the expectation. The Chinese government’s continued effort to promote the innovation and upgrading of secondary industry in the past was somewhat effective, and the CO_2_ emission intensity of secondary industry reduced from 3.87 kg/Yuan in 2008 to 2.22 kg/Yuan in 2018, but the total CO_2_ emission of second industry is still in the ascendant. Thus, it is necessary for China to implement more industrial upgrading policies.

ULGUE is influenced positively and significantly by technical progress level (*p* < 0.05). Technological progress is a critical factor in improving the environment [53]. Technical progress can be achieved through technology imports and independent innovation [54]. Only with independent innovation can China elevate and adjust its industrial structure. On the basis of digesting imported advanced technology from abroad, China needs to develop a series of innovative technologies to improve ULGUE.

The regression coefficient of the urban population density is significantly positive (*p* < 0.01), which implies that the urban population aggregation effect has positive impacts on ULGUE in China. However, that does not mean that the blind pursuit of urban population aggregation is wise. Currently, large numbers of people began to urban agglomeration in China [55]. The urban population density should stay in a certain range to guarantee the healthy development of cities.

Opening up a level playing field has a positive role in the promotion of ULGUE, but it is not significant. Opening up had a remarkably good effect on ULGUE, such as introducing advanced production technologies and management ideas; however, the export trade in many provinces of China is characterised by the extensive growth pattern. These two factors counteract each other, which may make the impact of opening up level on ULGUE unremarkable.

## 4. Conclusions

Due to limited land resources, the current rapid process of urbanisation and extensive utilisation is not a sustainable urban development model [56]. The green development of urban land lays great stress on both low resource consumption and low environmental impact [49]. This paper empirically studied the ULGUE in China, aiming to provide a sound assessment of ULGUE, and to ensure that the use of land and the development of our society is sustainable.

The article first evaluated the ULGUE in 30 provinces of China using the EBM model with undesirable outputs, and it found that the ULGUE showed significant spatial differences among provinces. Based on the spatiotemporal characteristics and research, the national mean level of ULGUE is shown to be fluctuating between 0.722 and 0.78, ULGUE shows a significantly positive spatial autocorrelation, and the spatial homogeneity was more significant than the spatial heterogeneity for ULGUE. In the end, the paper empirically analysed the driving factors of ULGUE using the SDM method. It found that economic development level, technical progress level, and urban population density have a significant impact on ULGUE, while the higher the proportion of the secondary industry of GDP is, the lower the level of ULGUE.

Based on the research results, the following countermeasures are put forward: (1) The government needs to continue playing a key role in policy regulation and imposing strict controls on urban incremental land. It needs to increase the support of land green development and strictly enforce enterprise emission standards and environmental protection access thresholds to avoid environmental degradation and urban sprawl. (2) It needs to exploit intensive land use potential fully, taking advantage of technical progress. On one hand, extending outward to import, adopt, and absorb foreign technology is necessary. More importantly, China should increase the input of R&D to develop technological innovation and re-innovation after digesting the introduced technology. (3) Promoting the green economic transformation of three industrial sectors is necessary, accelerating the development of labour-intensive industries to knowledge- and technology-intensive industries, promoting innovation and industrial upgrading, and advancing high-quality growth of manufacturing. (4) Accelerating the investment of human capital is necessary to support enterprises and private actors in providing vocational education. If cities have the characteristic of both high density and large-scale population, city governments should increase the requirements for rural residents to obtain hukous in large cities [57], and avoid the excessive growth of population inflow.

However, this study has limitations in terms of breadth and depth, so there is still space for improvement. First, in view of the evaluation index system, the undesirable output only included CO_2_, yet lacked other types of air and water pollution. Therefore, we suggest the NBSC improve the statistical work on air and water pollution to support a more sophisticated evaluation index system of ULGUE. Secondly, this study, based on the nationwide data, empirically researched the driving mechanism of ULGUE by adopting a reasonable spatial econometrics approach. Therefore, in order to yield more comprehensive results, future research should consider the regional heterogeneity and conduct empirical studies in each district. Furthermore, this paper measured the ULGUE based on the provincial administrative regions. Following research can expand the scale of the geographical unit to subordinate ones. In conclusion, future research can investigate the ULGUE of prefecture-level cities and reveal their spatial and temporal differences, so as to offer meaningful results for sustainable urban development.

## Figures and Tables

**Figure 1 ijerph-19-10748-f001:**
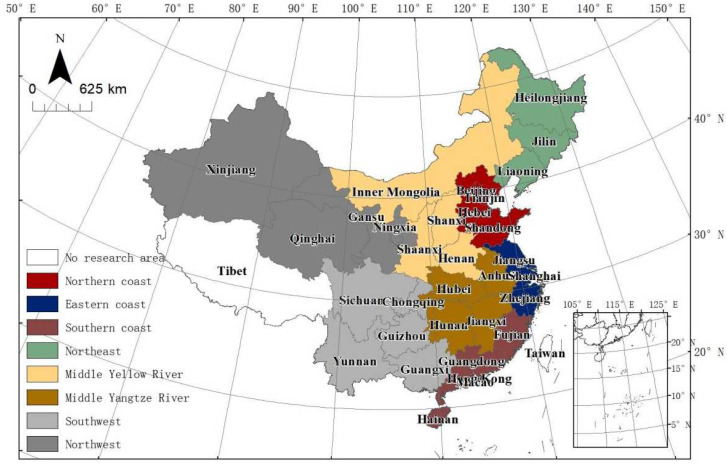
The schematic diagram of eight economic zones of China.

**Figure 2 ijerph-19-10748-f002:**
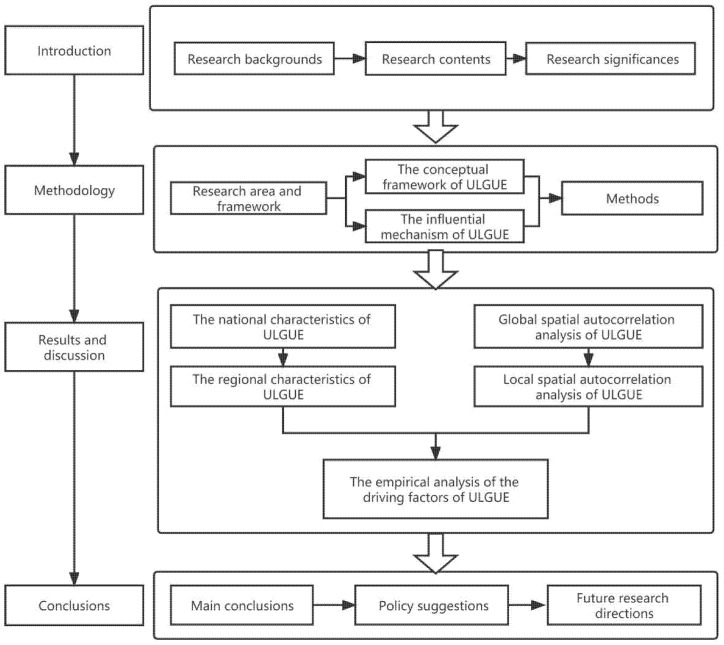
The framework of the empirical research of ULGUE in China.

**Figure 3 ijerph-19-10748-f003:**
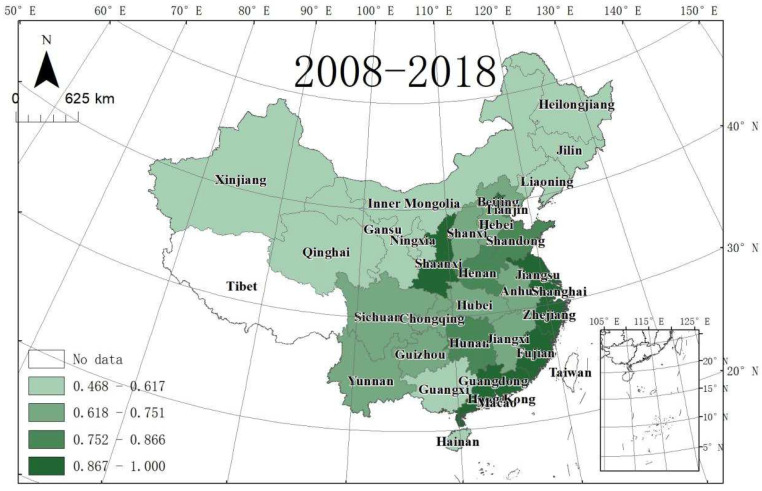
Average ULGLE values in 30 Chinese provinces (2008–2018).

**Figure 4 ijerph-19-10748-f004:**
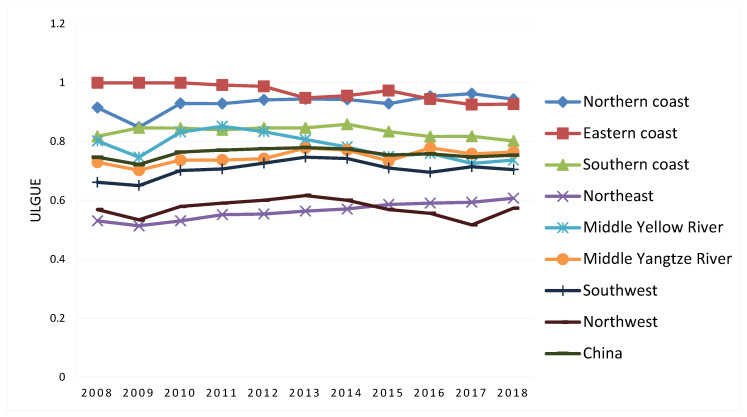
Evolutionary trend of ULGLE from 2008 to 2018.

**Figure 5 ijerph-19-10748-f005:**
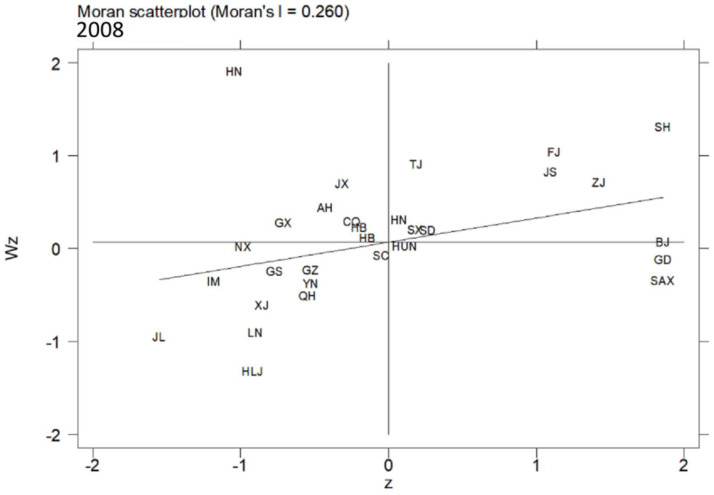
MSP map of ULGLE in 30 provinces in 2008.

**Figure 6 ijerph-19-10748-f006:**
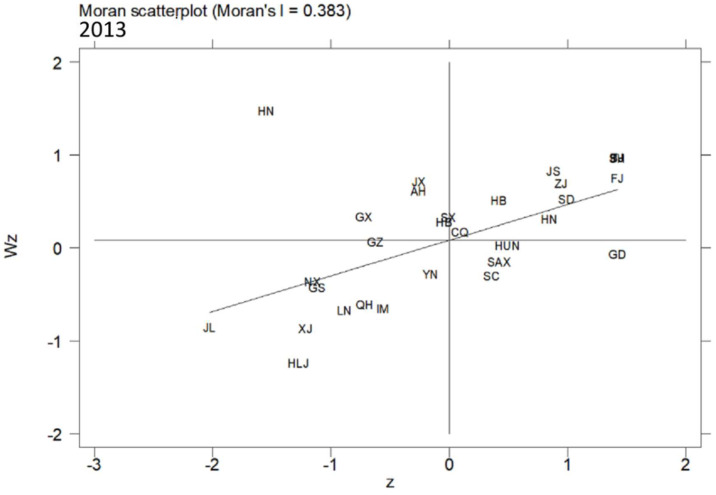
MSP map of ULGLE in 30 provinces in 2013.

**Figure 7 ijerph-19-10748-f007:**
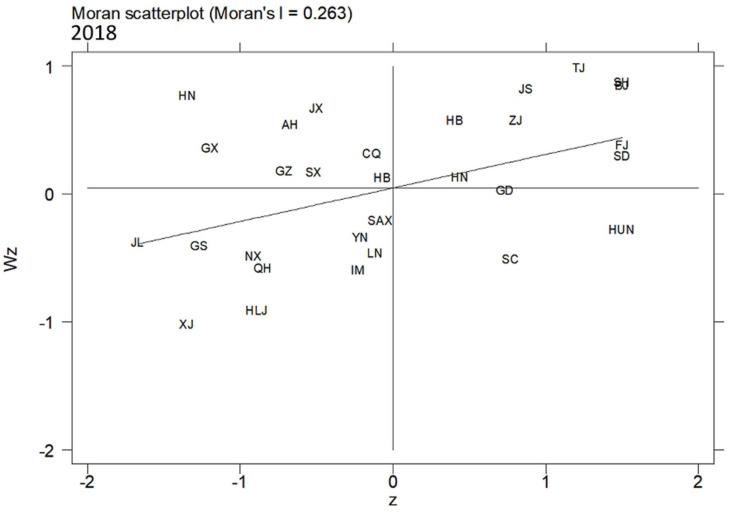
MSP map of ULGLE in 30 provinces in 2018.

**Figure 8 ijerph-19-10748-f008:**
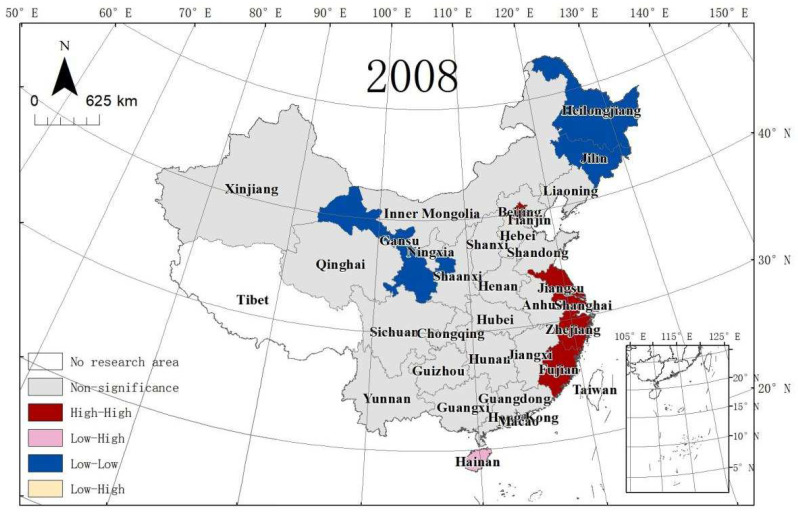
LISA of ULGLE in 30 provinces in 2008.

**Figure 9 ijerph-19-10748-f009:**
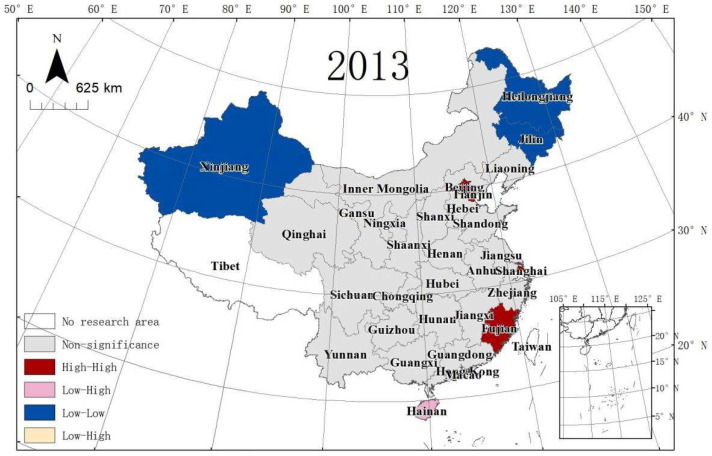
LISA of ULGLE in 30 provinces in 2013.

**Figure 10 ijerph-19-10748-f010:**
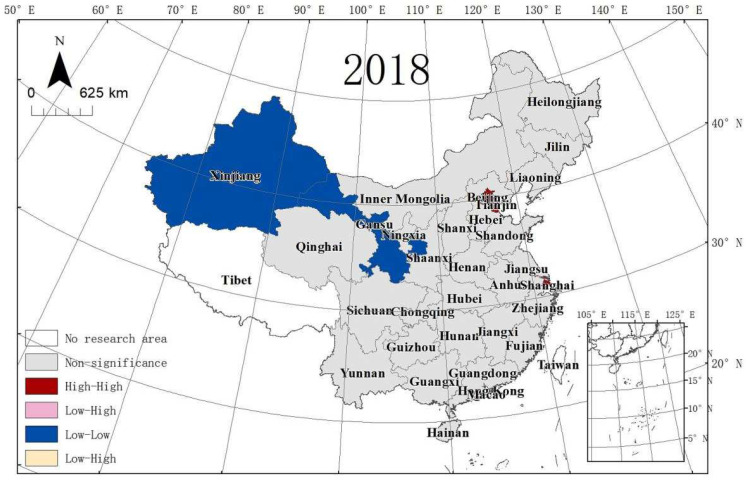
LISA of ULGLE in 30 provinces in 2018.

**Table 1 ijerph-19-10748-t001:** ULGUE evaluation indicator system.

Primary Indicators	Secondary Indicators	Std. Dev.	Max	Min
Inputs	Urban Capital stock (Unit: 10^8^ Yuan)	52,880	298,431	5141
	Urban employees (unit: 10^4^ persons)	853	5769	83
	Urban construction land (unit: 10^4^ mu)	1094	5577	109
Desired outputs	Added value of the second and the third industries (unit: RMB 10^8^ Yuan)	14,523	81,470	793
Undesired outputs	CO_2_ emissions of the second and the third industries (unit: 10^6^ tons)	189	869	23

**Table 3 ijerph-19-10748-t003:** The results of Wald and LR tests.

	Fixed Effects	Random Effects
Wald test spatial lag	87.55 ***	75.18 ***
Likelihood ratio test spatial lag	78.69 ***	70.76 ***
Wald test spatial error	70.90 ***	53.39 ***
Likelihood ratio test spatial error	74.50 ***	67.09 ***

*** represents *p* < 0.01.

**Table 4 ijerph-19-10748-t004:** The values of ULGUE for 30 provinces in China (2008–2018).

Regions	Provinces	2008	2009	2010	2011	2012	2013	2014	2015	2016	2017	2018	Mean
Northern coast	Beijing	1	1	1	1	1	1	1	1	1	1	1	1
	Tianjing	0.881	0.864	1	0.944	1	1	1	1	1	1	0.954	0.968
	Hebei	0.878	0.709	0.85	0.87	0.854	0.845	0.853	0.811	0.816	0.851	0.821	0.833
	Shandong	0.906	0.821	0.868	0.903	0.915	0.933	0.917	0.903	1	1	1	0.924
Eastern coast	Shanghai	1	1	1	1	1	1	1	1	1	1	1	1
	Jiangsu	1	1	1	1	1	0.917	0.941	1	0.901	0.879	0.897	0.958
	Zhejiang	1	1	1	0.975	0.964	0.926	0.928	0.921	0.933	0.899	0.886	0.948
Southern coast	Fujian	1	1	1	1	1	1	1	1	1	1	1	1
	Guangdong	1	1	1	1	1	1	1	1	0.903	0.855	0.874	0.967
	Hainan	0.454	0.541	0.538	0.521	0.541	0.54	0.577	0.503	0.549	0.599	0.533	0.536
Northeast	Liaoning	0.631	0.591	0.628	0.64	0.643	0.643	0.639	0.666	0.701	0.704	0.735	0.656
	Jilin	0.476	0.451	0.46	0.482	0.481	0.466	0.469	0.465	0.448	0.465	0.48	0.468
	Heilongjiang	0.486	0.5	0.504	0.535	0.538	0.582	0.606	0.629	0.625	0.612	0.608	0.566
Middle Yellow River	Shanxi	0.839	0.711	0.776	0.798	0.738	0.779	0.767	0.715	0.663	0.659	0.669	0.738
	Inner Mongolia	0.676	0.623	0.683	0.738	0.734	0.693	0.641	0.674	0.71	0.672	0.716	0.687
	Henan	0.877	0.803	0.874	0.871	0.863	0.911	0.89	0.846	0.851	0.815	0.826	0.857
	Shaanxi	0.815	0.85	1	1	1	0.846	0.829	0.765	0.815	0.759	0.741	0.856
Middle Yangtze River	Anhui	0.74	0.689	0.698	0.742	0.731	0.74	0.725	0.689	0.662	0.65	0.644	0.701
	Jiangxi	0.701	0.705	0.756	0.731	0.713	0.74	0.742	0.68	0.706	0.67	0.672	0.711
	Hubei	0.7	0.682	0.684	0.675	0.686	0.773	0.743	0.735	0.746	0.712	0.743	0.716
	Hunan	0.779	0.735	0.808	0.804	0.839	0.857	0.873	0.831	1	1	1	0.866
Southwest	Guangxi	0.579	0.618	0.637	0.638	0.634	0.668	0.676	0.634	0.579	0.569	0.558	0.617
	Chongqing	0.738	0.702	0.762	0.731	0.809	0.794	0.78	0.695	0.767	0.753	0.731	0.751
	Sichuan	0.734	0.715	0.788	0.818	0.828	0.835	0.834	0.701	0.707	0.876	0.88	0.792
	Guizhou	0.628	0.597	0.682	0.692	0.665	0.683	0.706	0.68	0.669	0.649	0.637	0.663
	Yunnan	0.629	0.621	0.639	0.656	0.698	0.756	0.717	0.838	0.759	0.726	0.719	0.705
Northwest	Gansu	0.553	0.547	0.559	0.58	0.595	0.607	0.593	0.575	0.546	0.523	0.546	0.566
	Qinghai	0.649	0.621	0.682	0.705	0.715	0.669	0.674	0.653	0.628	0.619	0.614	0.657
	Ningxia	0.544	0.455	0.537	0.526	0.542	0.601	0.556	0.503	0.517	0.505	0.604	0.535
	Xinjiang	0.53	0.514	0.542	0.553	0.552	0.592	0.58	0.546	0.533	0.421	0.532	0.536
Mean value of all provinces		0.747	0.722	0.765	0.771	0.776	0.780	0.775	0.755	0.758	0.748	0.754	0.759

**Table 5 ijerph-19-10748-t005:** ULGUE values for eight Chinese regions (2008–2018).

Regions	2008	2009	2010	2011	2012	2013	2014	2015	2016	2017	2018	Mean
Northern coast	0.916	0.849	0.930	0.929	0.942	0.945	0.943	0.929	0.954	0.963	0.944	0.931
Eastern coast	1	1	1	0.992	0.988	0.948	0.956	0.974	0.945	0.926	0.928	0.969
Southern coast	0.818	0.847	0.846	0.840	0.847	0.847	0.859	0.834	0.817	0.818	0.802	0.834
Northeast	0.531	0.514	0.531	0.552	0.554	0.564	0.571	0.587	0.591	0.594	0.608	0.563
Middle Yellow River	0.802	0.747	0.833	0.852	0.834	0.807	0.782	0.750	0.760	0.726	0.738	0.785
Middle Yangtze River	0.730	0.703	0.737	0.738	0.742	0.778	0.771	0.734	0.779	0.758	0.765	0.748
Southwest	0.662	0.651	0.702	0.707	0.727	0.747	0.743	0.710	0.696	0.715	0.705	0.706
Northwest	0.569	0.534	0.580	0.591	0.601	0.617	0.601	0.569	0.556	0.517	0.574	0.574

**Table 6 ijerph-19-10748-t006:** Value of Global Moran’s *I* of provincial ULGLE in China (2008–2018).

Year	Global Moran’s *I*	Z-Score	*p*-Value
2008	0.416 ***	3.603	0.000
2009	0.432 ***	3.749	0.000
2010	0.386 ***	3.364	0.001
2011	0.324 ***	2.863	0.004
2012	0.312 ***	2.769	0.006
2013	0.383 ***	3.350	0.001
2014	0.436 ***	3.771	0.000
2015	0.348 ***	3.072	0.002
2016	0.349 ***	2.929	0.003
2017	0.210 ***	3.977	0.002
2018	0.263 **	2.377	0.017

*** represents *p* < 0.01, ** represents *p* < 0.05.

**Table 7 ijerph-19-10748-t007:** The acronyms of 30 provinces in China.

Provinces	Acronyms	Provinces	Acronyms
Beijing	BJ	Henan	HN
Tianjin	TJ	Hubei	HB
Hebei	HB	Hunan	HUN
Shanxi	SX	Guangdong	GD
Inner Mongoria	IM	Guangxi	GX
Liaoning	LN	Hainan	HN
Jilin	JL	Chongqing	CQ
Heilongjiang	HLJ	Sichuan	SC
Shanghai	SH	Guizhou	GZ
Jiangsu	JS	Yunnan	YN
Zhejiang	ZJ	Shaanxi	SAX
Anhui	AH	Gansu	GS
Fujian	FJ	Qinghai	QH
Jiangxi	JX	Ningxia	NX
Shandong	SD	Xinjiang	XJ

**Table 8 ijerph-19-10748-t008:** The correlation test.

	LnULGUE	LnEDL	LnGR	LnIS	LnTPL	LnUPD	LnOUL
LnULGUE	1						
LnEDL	0.588 ***	1					
LnGR	0.130 **	−0.271 ***	1				
LnIS	0.663 ***	0.745 ***	−0.063	1			
LnTPL	0.022	−0.158 ***	0.222 ***	−0.086	1		
LnUPD	−0.676 ***	−0.343 ***	−0.333 ***	−0.484 ***	0.102 *	1	
LnOUP	0.529 ***	0.644 ***	−0.169 ***	0.623 ***	−0.164 ***	−0.563 ***	1

*** represents *p* < 0.01, ** represents *p* < 0.05, * represents *p* < 0.1.

**Table 9 ijerph-19-10748-t009:** The VIF test.

	LnEDL	LnGR	LnIS	LnTPL	LnUPD	LnOUL
VIF	2.81	2.16	2.67	1.1	2.46	1.60
1/VIF	0.356	0.462	0.375	0.912	0.407	0.625

**Table 10 ijerph-19-10748-t010:** The regression results of SDM.

Variables	Spatial Fixed-Effects	Time Fixed-Effects	Spatial and Time Fixed-Effects
InEDL	0.323 ***	0.286 ***	0.280 ***
InGR	−0.184 ***	−0.068	−0.075
InIS	0.075 **	−0.202 ***	−0.229 ***
InTPL	0.138 ***	0.076 **	0.067 **
InUPD	0.103 ***	0.053 **	0.048 **
InOUP	−0.083 ***	−0.004	0.007
W*InEDL	−0.535 ***	−0.159 *	−0.257 *
W*InGR	−0.073	0.1578 **	0.111
W*InIS	−0.164 *	0.615 ***	0.532 ***
W*InTPL	−0.033	−0.228 ***	−0.235 ***
W*InUPD	0.054	0.073 *	0.051
W*InOUP	0.150 ***	0.025	0.013
R-squared	0.621	0.202	0.122
Log likelihood	289.278	504.565	510.484

*** represents *p* < 0.01, ** represents *p* < 0.05, * represents *p* < 0.1.

## Data Availability

Data was obtained from China official national statistical database and China Emission Accounts and Datasets.

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
