# Peer review of "The Driving Mechanism of Urban Land Green Use Efficiency in China Based on the EBM Model with Undesirable Outputs and the Spatial Dubin Model"

_ijerph, 2022, doi:10.3390/ijerph191710748_

Round 1
Reviewer 1 Report
(1) The indicator system includes inputs, desired outputs and undesired outputs. The undesired outputs are CO2 emissions. Why not add some pollution variables like air pollution, water pollution, etc.?
(2) This paper is lack of conceptual framework and discussion.
(3) The introduction part can be further enriched to provide sufficient background and references.
Author Response
Reviewer 1
1.The indicator system includes inputs, desired outputs and undesired outputs. The undesired outputs are CO2 emissions. Why not add some pollution variables like air pollution, water pollution, etc.?
Respond:At present, we can only calculate the urban CO2 emissions (CO2 emissions of the secondary and tertiary industries). The current statistical yearbook does not provide air pollution and water pollution of tertiary industries, and only has the pollution data of the secondary. Therefore, the author only choose CO2.
2.This paper is lack of conceptual framework and discussion. Respond:Conceptual framework has added in line 112-122,and more discussion has added in lines 500 to 510.
3.The introduction part can be further enriched to provide sufficient background and references. Respond:More sufficient background and references has added in introduction of lines 26 to 35.
Reviewer 2 Report
General comments:
The manuscript, entitled "Spatiotemporal Characteristics and Driving Factors of Urban Land Green Use Efficiency in China Based on the EBM Model with Undesirable Outputs and the Spatial Dubin Model" by Liangen Zeng, applied an epsilon-based measure (EBM) model with undesirable outputs to evaluate the urban land green use efficiency (ULGUE) of 30 provinces in China from 2008 to 2018. There are some issues with this manuscript, mainly related to the readability and composition of the manuscript.
Please review the quality of your English throughout the manuscript.
I would like to see the method and some more recommendations of this study reported in the abstract section.
Specific comments:
Point 1: I recommend to the authors in the introduction part to add more citations in order to emphasize the need and originality of your research. Line 22-42.
Point 2: Move Figure 1 (Line 81) to the method section.
Point 3: In Figure 2 (Line 89), Figure 3 (Line 227), Figure 8 (Line 352), Figure 9 (Line 353) and Figure 10 (Line 354) add all map elements like grid and others also improve the layout.
Point 4: Line 427 in the conclusion and policy suggestion section no need for a citation.
Author Response
Reviewer 2
Point 1: I recommend to the authors in the introduction part to add more citations in order to emphasize the need and originality of your research. Line 86-90. Respond:More sufficient background and references has added in introduction in lines s 26 to 35.
Point 2: Move Figure 1 (Line 81) to the method section. Respond:the author has changed that in line 107
Point 3: In Figure 2 (Line 89), Figure 3 (Line 227), Figure 8 (Line 352), Figure 9 (Line 353) and Figure 10 (Line 354) add all map elements like grid and others also improve the layout. Respond:the author has added in the full figures.
Point 4: Line 427 in the conclusion and policy suggestion section no need for a citation. Respond:which has been changed.
Reviewer 3 Report
Congratulations to the authors for the work done.
The presented work is novel and deals with a tremendously interesting and problematic topic in China and other countries.
It presents minimal things in my opinion that should be improved:
1. The abstract is more of a statement of objectives than a summary of all the parts of the article. There is a lack of explanation of the methodology and the conclusions are not detected. It does not cover all parts. It is recommended to structure it better.
2. Figures 5, 6 and 7, result of Moran's I show poor graphic resolution. They are not seen clearly. Their resolution should be improved as they are not clearly understood. Furthermore, they are not easy to interpret. The content of these figures should be explained in more detail.
3. The text has an excess of acronyms and variables, making it difficult to read. Reading is not smooth. It is recommended to review the content to improve the writing.
Author Response
Reviewer 3
1.The abstract is more of a statement of objectives than a summary of all the parts of the article. There is a lack of explanation of the methodology and the conclusions are not detected. It does not cover all parts. It is recommended to structure it better. Respond: the author has changed that.
2.Figures 5, 6 and 7, result of Moran's I show poor graphic resolution. They are not seen clearly. Their resolution should be improved as they are not clearly understood. Furthermore, they are not easy to interpret. The content of these figures should be explained in more detail. Respond:the author has changed the all figures of MSP
3.The text has an excess of acronyms and variables, making it difficult to read. Reading is not smooth. It is recommended to review the content to improve the writing. Respond:the author has deleted some abbreviations, such as “DUM”, “LR” , and “L-L value”
Round 2
Reviewer 1 Report
There are still some problems to be revised before it can be published, otherwise I will suggest rejection.
(1) The structure of this paper should be improved, especially section 4 and section 5. I suggest section 5 should be incorporated in section 4 and the subtitle should be changed, for example “The driving mechanisms of …in China”
(2) section 5.1 and 5.2 should be removed. Authors should enrich section 3.3 and you can tell authors the steps of SDM. First we use multicolinearity tests ……, then we use wald and likelihood ratio tests. You donot need to list all the results. The more important is to let readers know your calculating steps.
(3) The driving mechanism is very important. However, authors just analyze the overall results. Because authors have analyzed the temporal trend and spatial pattern of ULGUE in section 4.1 and 4.2. Therefore, I suggest authors should add some empirical results about regional heterogeneity. China has four regions, coastal, central, western and northeastern, or three regions. You can demonstrate the results in a table and tell the readers the differences of driving mechanisms in four regions of China. Additionally, authors can also compare the results in different time periods.
(4) section 4.2, why authors use eight regions. To be consistent, I suggest authors analyze all the results by dividing China into three regions since your sample size is only 30 provinces.
Author Response
Dear Editor,
Thank you very much for your great comments and suggestions on our paper. We have modified the manuscript accordingly.
- The structure of this paper should be improved, especially section 4 and section 5. I suggest section 5 should be incorporated in section 4 and the subtitle should be changed, for example “The driving mechanisms of …in China”
Response:The author has combined the chapter 4 and 5 as the chapter 3””,The title was changed to “The driving mechanisms of Urban Land Green Use Efficiency in China Based on the EBM Model with Undesirable Outputs and the Spatial Dubin Model ”
- section 5.1 and 5.2 should be removed. Authors should enrich section 3.3 and you can tell authors the steps of SDM. First we use multicolinearity tests ……, then we use wald and likelihood ratio tests. You donot need to list all the results. The more important is to let readers know your calculating steps.
Response: The author has enriched the section 3.3 and added the introduction of the SEM, SLM models in lines 236 to 255, and performed Wald and LR tests for selection the more reasonable regression analysis method in lines 266 to 268
- The driving mechanism is very important. However, authors just analyze the overall results. Because authors have analyzed the temporal trend and spatial pattern of ULGUE in section 4.1 and 4.2. Therefore, I suggest authors should add some empirical results about regional heterogeneity. China has four regions, coastal, central, western and northeastern, or three regions. You can demonstrate the results in a table and tell the readers the differences of driving mechanisms in four regions of China. Additionally, authors can also compare the results in different time periods.
Response: The premise of using the spatial econometric model is that there is a strong spatial relationship for dependent variable. The author has made the Spatial autocorrelation analysis of ULGUE in the whole county of 30 provinces, in the Eastern region of 11 provinces, in the Central region of 8 provinces, and in the Western region of 11 provinces, respectively. However, the tests found that the p-values are significance of 5% at the national level from 2008 to 2018, and the p-value is significant at the central regions from 2008 to 2015, but which are not significant at the eastern region and western region from the whole study years.
Therefore, the spatial econometric model cannot be used for the empirical research in the East, central and western regions. It only can use the non spatial econometric models, such as Tobit model to empirically research the driving factors of ULGUE in the East, central and western regions. However, the spatial econometric model (SDM) is used in the whole country, while Tobit, a non spatial econometric model, are used in the East, middle and West. This combination is not very suitable.
Therefore, this paper suggests that the research of the driving mechanism of ULGUE by a more reasonable spatial econometrics approach based on the nationwide data, and the heterogeneity analysis of the East, middle and West should not be considered in this study. The heterogeneity analysis of the East, West and East should be the future research directions by using the non spatial econometrics model, which has been added in lines 525 to 529.
- section 4.2, why authors use eight regions. To be consistent, I suggest authors analyze all the results by dividing China into three regions since your sample size is only 30 provinces.
Response: The concept of the eight regions was the proposed by the State Council of China in 2005, which considers the geographical location and regional economic development level ( https://baike.baidu.com/item/%E5%85%AB%E5%A4%A7%E7%BB%8F%E6%B5%8E%E5%8C%BA/7881543#viewPageContent ). It is a more reasonable regional classification than the traditional method (East, Central, and West), Therefore, this paper uses the eight regions for the regional characteristics analysis.
The manuscript has been resubmitted to your journal now.
Regards
Liangen Zeng
